# Uncovering the Mechanism of Online-Learning Stress of College Students

**Enuo Wang** [1,*] **and Xueyao Zhang** [2]

1    School of Foreign Languages, University of Shanghai for Science and Technology, Shanghai 200093, China
2    School of Foreign Studies, University of International Business and Economics, Beijing 100029, China; xueyao_zhang@126.com
*    Correspondence: petranuo@sina.com

**Abstract:** Online-learning stress poses a significant challenge to the sustainability of higher education. The present study employs mixed methods to propose a conceptual process model that depicts the mechanism of online-learning stress of college students. The result of the qualitative study indicates 11 influential factors of online-learning stress, 10 manifestations of online-learning stress (OS), and three learning performance outcomes of OS (LP) through in-depth interviews with 15 college students. The result of a quantitative study on 159 online surveys implies that the influential factors of online-learning stress could be further categorized into learner competence and commitment (LC), course design reasonability (CD), and social support (SS). In addition, the results of the structural equation model (SEM) confirm the negative impact of LC and CD on OS, as well as OS on LP. However, the negative effect of SS on OS is unsupported. The study contributes to both OS theory development and online-learning and teaching in higher education.

**Keywords:** online-learning stress; online-learning; digital education; higher education sustainability

## 1. Introduction

Higher education has entered a new era of digitalization. In the field of digital education, the application of online learning has become a trend since the beginning of the 21st century [1]. Online learning provides learners with the opportunity to engage in virtual classrooms through the internet, using computers or smartphones. Online learning benefits college students around the world by reducing the learning cost and improving the convenience of learning without the constraints of time and space [2,3]. The breakout of the COVID-19 pandemic sped up the transition from traditional offline learning to online learning [4,5]. According to the report released by the International Association of Universities in Paris, at the height of the lockdown in Europe, 85% of European universities reported that courses were delivered through online learning [6]. Though online learning has some positive effects on college students, the unexpected rapid switch from offline to online still brings about many mental issues to college students [7]. Among the psychological problems, online-learning stress has become one of the most common psychological issues for college students, which deserves to be brought to the attention of the wider community [8].

The existing academic works on college students' stress can be generally grouped into two parts, namely, learning stress and living stress. Specifically, learning stress refers to stressors strongly related to students' academic events, such as the stress of achieving good grades and taking courses. Living stress, on the contrary, refers to students' daily stressors that are less relevant to their studies, such as the stress of maintaining friendships, the stress of paying tuition, etc. We found that many scholars in higher education prefer to concentrate on learning stress [9,10]. It is not only because learning stress is extremely important to college students' academic performances but also because it is inspiring for

teachers to improve their teaching methods and course designs [11,12]. Normally, studies on the learning stress of college students focus on their stress under different learning circumstances, e.g., the stress of taking a course [13], the stress of doing homework [14], the stress of taking exams, and the stress of learning a specific major [15]. Recently, with the outbreak of the COVID-19 pandemic, some scholars have turned their attention to the online-learning stress. They separate online-learning stress from traditional learning stress (mainly referring to offline-learning stress) and present some new findings concerning learning stress [16]. However, the number of relevant articles remains limited.

Few scholars work on the definition and manifestations of online-learning stress. To define online-learning stress, scholars tend to borrow concepts from perceived stress and academic stress [17]. In the study by Liu et al. [18], forms of online-learning stress can be feeling of anxiety, attention distraction, and worry, according to interviews with college students. Most of the existing studies try to explore the influential factors of online-learning stress. We summarize those factors into four types: student-relevant factors, teacher-relevant factors, technology-relevant factors, and society-relevant factors. For example, Al-Kumaim et al. [19] present in their article that personal factors, technical factors, and social-environmental factors could lead to mental issues in college students during the outbreak of COVID-19. Jung et al. [20] list self-efficacy, instructional design, technology use, and online-collaborative processes. Lazarevic and Bentz [21] report the time commitment of learners, accessibility to learning materials, the social environment, and the expectation of family and friends as influential factors. The influential factors can be either positive or negative to the online-learning stress of college students. Duraku and Hoxha [22] identified that the impact of the emotional support of teachers on stress is positive, while the effects of students' motivation on learning, students' self-control, family members' interference, and workload of the course on online-learning stress are negative. Lastly, there are few studies that discuss the impact of online-learning stress. The impact of online-learning stress remains controversial. Some researchers agree that online-learning stress can lead to negative outcomes. Online-learning stress can decrease the level of academic self-efficacy and academic hope [23]. However, some scholars compare the online-learning stress with offline-learning stress and surprisingly find that the stress perceived online can also have some positive impacts [24]. Kumalasari et al. [17] suggest the mediation effect of academic resilience that could help students overcome stress and achieve learning satisfaction.

By reviewing the previous literature, we conclude that most researchers are only concerned with influential factors of online-learning stress and pay less attention to the manifestations and effects of the stress, not to mention the overall online-learning stress mechanism. In addition, their research methods are mostly homogeneous. Scholars conduct either purely explorative qualitative research or purely confirmatory quantitative research.

Based on both the practical and theoretical necessity of exploring online-learning stress indicated above, we decided to work on the mechanism of online-learning stress in our study. More specifically, we try the answer three questions:

Q1: What are the manifestations of online-learning stress?

Q2: What factors cause the online-learning stress?

Q3: What are the possible effects caused by online-learning stress on college students' academic lives?

We structure the study as follows to arrive at the answers to our research questions: firstly, we overview the general context for the focal topic in our study to prove the theoretical and practical significance of the present study. Then, to demonstrate the academic rigor of the study, we carefully cover the data collection and analysis process in our mixed-method study. The results of our qualitative study present all the potential key concepts that could build up the mechanism of online-learning stress. We further test the relationship among those key concepts and finally create the refined mechanism of online-learning stress as the result of a quantitative study. Next, we discuss our findings by relating them to the previous literature in order to establish a theoretical framework and further demon-

strate our contribution to the relevant field. Finally, the study finishes with a conclusion, presenting both theoretical and practical implications, limitations, and future directions.

## 2. Materials and Methods

The whole research procedure of the study was structured into two steps. Firstly, we conducted the explorative qualitative study, i.e., the in-depth interview and its analysis. During the in-depth interview, we identified and generalized 24 key concepts belonging to three main dimensions of online-learning stress, namely the influential factors of online-learning stress, online-learning stress' manifestations, and the impact of online-learning stress. Then, we conducted the explorative-confirmatory quantitative study, i.e., the online survey and its analysis. With the statistical methods of SEM, we further refined and tested the online-learning stress mechanism by dividing 10 influential factors into three sub-dimensions and ruling out two key concepts within online-learning stress manifestations. In the next section, we present the materials and methods for each step.

### 2.1. Participants

For the in-depth interview, 15 college students in China were selected as our participants (see Table 1) via a nationwide online community platform. We took the sampling method of purposive sampling, which is frequently used in qualitative studies [25]. Following the criteria of "selecting information-rich cases for study in depth" [26], all the college students participating in our interview reported extensive experience in online learning and varying degrees of online-learning stress. In addition, to gain a more general understanding of students' online-learning stress, we purposefully selected students from 15 different colleges in 15 different cities of China, ranging from top universities located in Beijing and Shanghai to basic public universities located in third-tier cities of southwest China.

**Table 1.** Demographic information of interview participants.

| No. | Gender | Grade | Major | Online-Learning Experience |
|---|---|---|---|---|
| P-1 | Female | 3rd year, undergraduate | Special Education | 2.5 years |
| P-2 | Female | 3rd year, undergraduate | Media Studies | 3 years |
| P-3 | Female | 2nd year, undergraduate | Geoscience | 3 years |
| P-4 | Female | 1st year, graduate | Communication Studies | 2 years |
| P-5 | Male | 5th year, PhD | World History | 2 years |
| P-6 | Female | 1st year, graduate | English Education | 2 years |
| P-7 | Male | 2nd year, graduate | Film Studies | 2 years |
| P-8 | Female | 4th year, undergraduate | Fashion Design | 1.5 years |
| P-9 | Female | 3rd year, undergraduate | Social Work | 2 years |
| P-10 | Female | 4th year, undergraduate | Law | 2.5 years |
| P-11 | Female | 4th year, undergraduate | Sports Management | 2 years |
| P-12 | Male | 3rd, undergraduate | Accounting | 2 years |
| P-13 | Female | 2nd year, undergraduate | Politics | 2 years |
| P-14 | Male | 3rd year, PhD | Marketing | 2 years |
| P-15 | Male | 2nd year, graduate | Computer Science | 2.5 years |

Taking the methods of snowball sampling, we were able to reach 165 Chinese college students for our online survey. More specifically, we first posted our research theme and the link to our online questionnaire on a Chinese social media platform (Douban) with a large number of college users across China. Then, those initial participants (43 college students) were invited to forward the online questionnaire to their college friends. In the end, 165 students from 38 colleges in 23 provinces were recruited to attend our online survey. Among the 165 questionnaires returned, 159 questionnaires were valid. Demographic information of the valid participants can be seen in Table 2.



**Table 2.** Demographic information of online survey participants.

| Demographic Information | Frequency (n = 159) | Percent (%) |
|---|---|---|
| Gender | | |
| Male | 65 | 41 |
| Female | 94 | 59 |
| Grade | | |
| 1st year, undergraduate | 44 | 28 |
| 2nd year, undergraduate | 13 | 8 |
| 3rd year, undergraduate | 30 | 19 |
| 4th year, undergraduate | 13 | 8 |
| 1st year, graduate | 25 | 16 |
| 2nd year, graduate | 10 | 6 |
| 3rd year, graduate | 19 | 12 |
| Ph.D. student | 5 | 3 |
| Major | | |
| Language Studies | 20 | 13 |
| Finance | 9 | 6 |
| Engineering | 15 | 9 |
| Biology | 6 | 4 |
| Statistics | 13 | 8 |
| Computer Science | 15 | 9 |
| Media Studies | 11 | 7 |
| Geography | 4 | 3 |
| Law | 17 | 11 |
| Education | 7 | 4 |
| Politics | 9 | 6 |
| Art | 5 | 3 |
| Business Studies | 8 | 5 |
| Psychology | 4 | 3 |
| Sociology | 6 | 4 |
| Other | 10 | 6 |
| Online-learning experience | | |
| less than 0.5 years | 12 | 8 |
| 0.5–1 year | 21 | 13 |
| 1–3 years | 81 | 51 |
| 3–5 years | 34 | 21 |
| more than 5 years | 11 | 7 |

*2.2. In-Depth Interview*

We first conducted explorative in-depth interviews online in order to identify the mechanism of the online-learning stress perceived by college students.

The in-depth interviews were semi-structured, and topics were arranged on the basis of our research questions, namely the stress of online learning itself and its manifestations, the factors that contribute to online-learning stress, and the impact of online-learning stress on college students in their academic life. Altogether, each interview contained around 15–18 questions and lasted 30–40 min regarding actual circumstances (see Appendix A). We recorded all the conversations during each interview and then transcribed them into text after each interview.

As for the ethical consideration, all the participants were informed about the purposes, potential risks, and benefits of the study before the interview. At the beginning of the interview, each participant verbally confirmed that their participations were anonymous and voluntary. During the interview, participants had the right to skip questions they did not want to answer. All the transcriptions of the in-depth interviews were permitted and confirmed once again by the participants after the interview. In addition, the interview outline obtained approval from the ethics committee of the authors' home university.

### 2.3. Online Survey

We also conducted an online survey to confirm the findings in interviews. We designed a questionnaire with 27 questions (see Appendix B) based on the three groups of key concepts (influential factors, online-learning stress and its manifestations, and impacts of online-learning stress in academic life) generated from the in-depth interviews. As for the measurements for online-learning stress, we additionally referred to the classic stress assessment tool Perceived Stress Scale (PSS) [27] and adapted it to the context of online learning. All the items were Likert-type and scored on a scale of 1 to 5 (in items except online-learning stress, 1 = totally agree, 5 = totally disagree; in items of online-learning stress, 1 = never, 5 = very often). All the questionnaires were distributed on an online survey platform for college students, and in the end, 159 valid questionnaires were collected, as mentioned above.

Participants attending the online survey were also informed about the purposes, potential risks, and benefits of the study on the front page of the questionnaire. The generation and analysis of data were conducted under the permission of the participants. The questionnaire applied in the survey was also checked and approved by the ethics committee of the authors' home university.

### 2.4. Data Analysis

Qualitative and quantitative methods were used for data analysis in our two-step study.

We applied an explorative qualitative method following the Qualitative Analysis Guide of Leuven (QUAGOL) [28] to analyze the data generated from explorative in-depth interviews. We went through all 10 stages of QUAGOL. During the analysis, we re-read all the transcripts of interviews, formed a brief abstract of key storylines of each interview, coded concrete expressions into concepts, analyzed the concepts, and grouped the key concepts into a general process model. Additionally, to enhance the reliability of the coding process, the two authors coded the transcripts separately and then communicated with each other to reach an agreement. An expert in the field of higher education was later invited to ensure the quality of coding.

With the data drawn from the online survey, we applied the method of structural equation modeling (SEM) to test and refine the process model generated from the qualitative analysis with SPSS Amos 26. Specifically, we grouped key concepts into components with the help of explorative factor analysis (EFA) and further refined the model through confirmatory factor analysis (CFA). Finally, we used path analysis to test the refined model and figure out the mechanism of online-learning stress of college students.

## 3. Results

### 3.1. Qualitative Study Results

In the following section, we generalized all the potential key concepts that could build up the mechanism of online-learning stress (influential factors, manifestations, and impacts) based on the explorative in-depth interviews.

### 3.1.1. Online-Learning Stress of College Students

Participants in the in-depth interviews used diverse expressions to explain their perception of online learning stress. The manifestations of their OS could be summarized as follows:

- Feeling of attention distraction (OS1)

College students under the situation of attention distraction were not able to concentrate on their course. Some participants complained that they always found themselves undergoing "unconscious mind-wandering" during the period of online learning. P-5 confirmed this form of OS, "*What I felt stressed was that I always ended up with finding myself distracted and didn't hear any of the key points from the teacher.*" P-12 had the same feeling, "*I*

*was not able to take down any notes since I was pretend to be paying attention to the course. But in fact, I knew I was not concentrating. That's terrible.*"

- The feeling of an inability to control important things in learning (OS2)

Some participants expressed stressed feelings as a high level of losing control. More specifically, they felt that they were even unable to organize and complete the most important issues in their academic life. P-1 remembered the first pair of days when undergoing online learning courses at home, "*I was not the boss of my study anymore. In contrast, I felt like I was the employee, or even the slave of the time. I studied what time pushed me to do.*" P-8 felt the same during the period of online learning. She tried hard to work on her bachelor paper; however, she noticed that it was hard for her to "*plug those writing-stuffs in her life*".

- The feeling of mental tension for no reason (OS3)

Participants reported a state of mental tension during the online learning period. They struggled to identify the direct triggers for the durable mental tension. P-13 felt nervous all the time, even though she was "*watching chill vlogs on social media platforms*". Similarly, P-15 struggled with the mental tension at night: "*I felt that my brain tightened up even when I was about to fall asleep. It became a type of sleep disorder for me*".

- The feeling of losing confidence in dealing with personal issues (OS4)

The feeling of being unconfident was mentioned by many participants. Most of them recalled their offline learning circumstances, in which they did not frequently question themselves while dealing with personal issues. However, in the online learning period, they tended to doubt that they would "*fail in dealing with any things, even personal learning issues (P-2, P-7, P-10)*".

- The feeling of unwilling direction in which things are going (OS5)

This was a feeling that things always failed to go as people wanted them to. P-4 described this feeling as "*being the enemy of the world*". P-14 felt "*unlucky with everything since results always turned out to be the unexpected one*" while learning online.

- Feeling of inability to follow the daily routine (OS6)

This kind of feeling occurred particularly in college students who had to attend early 8:00 a.m. online courses. P-6 admitted that she never succeeded in taking the online course at 8:00 a.m.: "*It was weird that I felt that I was not able to cope with daily-routine-activities that I have to do anymore.*" P-11 could understand her most: "*I felt that the daily plan was no longer useful.*"

- The feeling of an inability to tackle the irritating things in studies (OS7)

Participants confirmed that the irritative issues in studies were increasingly uncontrollable. In the past, they felt that, though they confronted obstacles while learning, they did not regard those obstacles as irreconcilable. However, during the online learning period, the feeling of taking control of the irritating things was weakened. College students felt that they were "*defeated by difficulties in some courses (P-3)*" and expressed a feeling of "*lying flat on the ground and doing nothing (in Chinese 躺平, P-1, P-4, P-8, P-9)*" while facing difficulties while learning online. This was a severe stress-out situation.

- The feeling of an inability to control anything in life (OS8)

This was a feeling of totally losing control, which was also regarded as a more severe stress-out situation than the inability to control important things in learning. P-10 looked back to those online-learning days and was shocked to find that she could hardly remember a thing. "*Muddleheaded*" was the word mentioned several times to describe this feeling (P-12, P-13).

- The feeling of irritation and anger toward studying (OS9)

The common feeling of irritation and anger occurred frequently. College students found it easier to get angry while taking online courses. Even the tone of the teacher could irritate them. "*We shared a common feeling of dissatisfaction towards courses and teachers, we always spitted online in we-chat groups while having online courses. Anything related to study had the potential to make us angry*", reported P-3.

- The feeling of an inability to complete multiple tasks (OS10)

The ability to complete multiple tasks within a certain time period was once a basic competence of college students. However, some felt that they lost this basic competence of multi-task complement. P-2 argued that she always found that "*difficulties were piling up and you would not even try to overcome them*". P-7 shared the same feeling: "*I used to be efficient, however, during the online-learning period, I started to procrastinate tasks. I felt that I was no longer efficient in doing study-related-things simultaneously. The only thing I was confident to complete at the same time was taking online-course while swiping Weibo (a Chinese social media platform).*"

3.1.2. Influential Factors of Online-Learning Stress

- Interaction with classmates (CI)

Most participants agreed that there was a lack of communication among classmates both during and after class. For some participants, a lack of interaction with classmates could strengthen their feeling of online-learning stress (P-2, P-10, P-13). In addition, some regarded the interaction with classmates as stressful (P-9, P-15).

- Interaction with teachers (TI)

Interaction with teachers was also recognized as a possible influential factor. Just like the interaction with classmates, their functions on online-learning stress were also double-edged. Some hated interaction with teachers, especially those conducting "interactions in form". P-14 further explained, in his opinion, courses starting with "I would like to pick up one student to answer my question" or ending up with "Do you have further questions" were meaningless. Instead, it added online-learning pressure. However, more participants supported the interaction with teachers. P-1 felt that she "*did not comprehend any knowledge unless interacting with teachers during the class*".

- Emotional support during the course (ES)

Emotional support was a key concept mentioned by P-7 and P-13. They argued that the feeling of online-learning stress would be more severe if the teacher of the course did not show sympathy to their students and neglected their psychological situation.

- Self-efficacy (SE)

Self-efficacy was an important learner characteristic stressed by several participants. Low self-efficacy made some participants less confident in facing changes in studies. P-13 defined herself as a "*low self-efficacy, insecure and weak*" person and further related her personal traits to her reasons for being stressed and depressed.

- Learning time commitment (LTC)

Some participants attributed their online-learning stress to less preparation for studies. More specifically, P-2 and P-9 expressed a feeling of guilt when they did not devote enough time to their studies. At the same time, being less prepared enhanced their feeling of inability to do homework and further tasks, as well as the possibility of getting angry with the completion of homework.

- Accessibility to combined multiple learning methods (MC)

During the online-learning period, different courses allowed different levels of learning methods combination. For example, some participants felt relaxed if they were able to combine offline learning methods while taking online courses. P-10 gave us an example of how her online-learning stress was greatly relieved since she was able to take down

notes on the real physical textbook, "*Writing down on the paper could greatly comfort me*", she concluded. Some of the participants shared the same tendency as P-10.

- Accessibility to combined online-learning devices (DC)

Learning devices were important for online learning. With the assistance of multiple online-learning devices, college students were able to separate the function of each device and thus enhanced their feeling of study efficiency. Browsing the transcripts of the interview, we noticed that most of the college students used more than one device. Normally, they used portable computers as a virtual classroom and then tablets as their study assistant. P-12 explained his stress with single online-learning devices: "*At the beginning, I only used PC for learning. Sometimes, the teaching platform broke down when running too many programs. To fix the problem, I sometimes took risk of losing all the notes on PC when I tried to shut down the PC. It was a really unwilling situation*".

- Reasonability of workload (WLR)

The unreasonable workload of online courses brought a high level of tension feeling. P-9 pointed out that most of her feeling of tension, anger, and depression were caused by unreasonably huge workloads. Other participants added that an unreasonable workload within a short time limit would "*drive people crazy (P-3, P-10)*".

- Reasonability of course difficulty (CDR)

The difficulty of online courses was expected to be moderate. Courses that were too easy were regarded as "time-wasting courses (水课)". Meanwhile, those too hard were regarded as "energy-consuming courses". Either type of course brought stress to students. A "good course" should be "*stuck in the middle (P-15)*", which could provide a "*sense of achievement and relief at the same time (P-4, P-8)*".

- Timely feedback by the teacher of the course (TF)

Teachers of the online course should provide timely feedback to students. In other words, teachers of online courses should be responsible and efficient in helping a student solve their learning problems. P-11 said she felt stressed when she sent her teacher a message with questions via WeChat and waited for two days to receive feedback.

- Teacher's proficiency in professional knowledge of the course (PPK)

Some of our participants argued that the feeling of stress depended largely on their teachers, especially on their proficiency in professional knowledge. P-13 felt relaxed when she noticed that the teacher of the course was "*well-experienced with respectful major expertise*". Their proficiency could decrease the students' feelings of uncertainty while they were learning and having difficulties with specific questions.

3.1.3. Impact of Online Learning Stress in Academic Life

- Attitudes changes (AC)

Many college students reported their emotional changes toward learning. After taking online courses and experiencing all forms of stress, most participants found themselves "*losing interest in taking course and even in learning anything (P-4)*". P-14 thought that "*courses were not so attractive as before*". Despite negative attitudes changes towards studies, P-15 found, "*some courses were more attractive via online-teaching, for example English.*"

- Cognitive changes (CC)

After the period of online learning and stress-out situations, most participants confessed that they had "*poor knowledge*" concerning their major (P-6, P-8). P-11 provided us with a typical example, "*My major required swimming competence, so, we had to take swimming course online. During the whole semester, I had never been to a swimming pool, not to mention the comprehension of swimming in the water.*" From this case, it was clear that cognitive performance and changes in studies were also major- and course-related. However, for a few

participants, their comprehension in specific courses (mathematics for P-14, English for P-15) was improved.

- Academic performance changes (AP)

Grades achieved during the online-learning period were reported to be the same (P-2, P-3, P-6, P-15) or worse (other participants) compared to the offline-learning semester. P-12 lowered his expectation towards academic performance: "*In those unusual times, a pass was sufficient for me.*"

### 3.1.4. Key Concept List and Conceptual Process Model

To conclude, we achieved the key concept list (see Table 3) according to the results of the qualitative study. Key concepts are also tagged for the use of further analysis.

**Table 3.** Key concept list.

| Dimension | Key Concept in Tag | Dimension | Key Concept in Tag | Dimension | Key Concept in Tag |
|---|---|---|---|---|---|
| Online-learning stress of college students | OS1 | Influential factors of online-learning stress | CI | Impact of online learning stress in academic life | AC |
| | OS2 | | TI | | |
| | OS3 | | ES | | |
| | OS4 | | SE | | |
| | OS5 | | LTC | | CC |
| | OS6 | | MC | | |
| | OS7 | | DC | | |
| | OS8 | | WLR | | |
| | OS9 | | CDR | | AP |
| | OS10 | | TF | | |
| | | | PPK | | |

Additionally, by referring to the contents of in-depth interviews, we formed a general process model implying the interrelationship between key concepts and dimensions (see Figure 1).

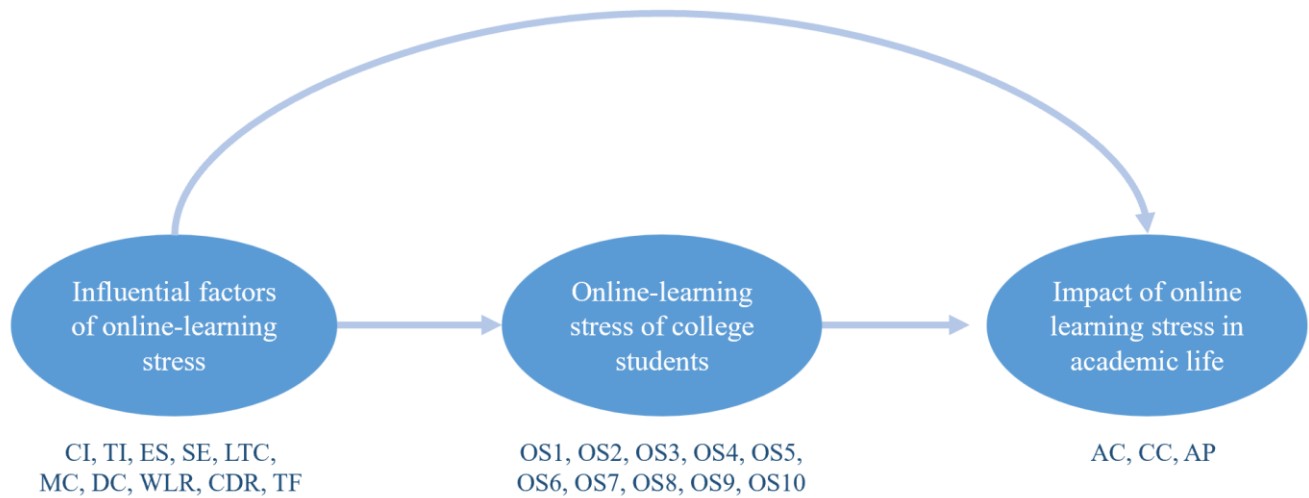

**Figure 1.** The conceptual process model of the online-learning stress mechanism.

### 3.2. Quantitative Study Result

#### 3.2.1. EFA Test

EFA was used as a fundamental tool in the exploration and validation of theories and measurements [29]. Based on the key concepts induced from the explorative in-depth

interviews, we undertook EFA analysis to identify the smallest number of hypothetical common factors (also called dimensions in the former section) with the assistance of SPSS 26. Those common factors could then better present and explain the inter-relationship among those key concepts (also called measured variables in EFA analysis) and, thus, the structure of the process model.

Firstly, we tested the appropriateness of our data for EFA. The Bartlett test indicated that the correlation matrix was not random, $\chi 2 = 2769.551$, $p < 0.001$. The result of KMO was 0.9, well above the criteria of 0.6.

As for the factor analysis, we conducted a principal components analysis (PCA) to decide the number of included factors. The result of PCA suggested that five factors should be retained in our structure model. Under the total variance explained, five rows of eigenvalues (each factor's value was greater than 1) took up 73.061% of the total extraction sums of squared loadings.

Then, we conducted factor rotation to interpret factor loadings (see Table 4).

**Table 4.** Rotated Factor Matrix *.

|  | Factor 1 | Factor 2 | Factor 3 | Factor 4 | Factor 5 |
|---|---|---|---|---|---|
| SE | −0.318 | 0.189 | 0.002 | 0.690 | 0.313 |
| LTC | −0.123 | 0.116 | 0.209 | 0.850 | 0.092 |
| MC | −0.128 | 0.088 | 0.187 | 0.777 | 0.307 |
| DC | −0.078 | 0.821 | −0.030 | −0.006 | 0.211 |
| WLR | −0.198 | 0.809 | 0.174 | 0.100 | 0.123 |
| CDR | −0.254 | 0.700 | 0.176 | 0.262 | 0.089 |
| CI | −0.160 | 0.159 | 0.805 | 0.077 | 0.296 |
| TI | −0.171 | 0.324 | 0.795 | 0.134 | 0.155 |
| ES | −0.113 | 0.271 | 0.720 | 0.229 | 0.117 |
| TF | −0.146 | 0.723 | 0.447 | 0.065 | 0.025 |
| PPK | −0.062 | 0.677 | 0.368 | 0.103 | 0.082 |
| OS1 | 0.706 | −0.097 | −0.172 | −0.196 | −0.109 |
| OS2 | 0.692 | −0.084 | −0.164 | −0.341 | −0.208 |
| OS3 | 0.788 | −0.004 | −0.106 | 0.156 | −0.159 |
| OS4 | 0.840 | 0.020 | −0.151 | −0.051 | −0.051 |
| OS5 | 0.846 | −0.152 | −0.120 | −0.033 | −0.041 |
| OS6 | 0.778 | −0.227 | 0.036 | −0.113 | −0.064 |
| OS7 | 0.811 | −0.122 | −0.075 | −0.174 | −0.150 |
| OS8 | 0.827 | −0.120 | −0.081 | −0.199 | −0.080 |
| OS9 | 0.845 | −0.214 | −0.028 | −0.096 | −0.065 |
| OS10 | 0.788 | −0.124 | −0.051 | −0.118 | −0.267 |
| AC | −0.191 | 0.111 | 0.311 | 0.148 | 0.789 |
| CC | −0.204 | 0.191 | 0.237 | 0.266 | 0.774 |
| AP | −0.257 | 0.219 | 0.069 | 0.296 | 0.777 |

* Extraction Method: Principal Axis Factoring; Rotation Method: Varimax with Kaiser Normalization; Rotation converged in three iterations.

Results above informed us with factors and their underlined key concepts (here, the measured variables): as expected, OS1-OS10 could be grouped into a common factor: online-learning stress and its manifestations (tagged as OS). In addition, AC, CC, and AP were in the same group of factors, the outcomes of online-learning stress of college students in terms of learning performance (tagged as LP). Moreover, DC, WLR, CDR, TF, PPK, CI, TI, ES, SE, LTC, and MC were classified under a common factor, respectively. These results inspired us to further group the influential factors of online-learning stress into three groups, namely, course design reasonability (tagged as CD), social support (tagged as SS), and learner competence and commitment (tagged as LC), respectively. The indicator of reliability, namely Cronbach's alpha coefficient was satisfied within each factor group: 0.785 (LC), 0.869(CD), 0.850 (SS), 0.947 (OS), and 0.873 (LP).

Given these results, the five-factor solution was accepted as the most adequate model to describe the mechanism of online-learning stress with our online-survey data so far.

### 3.2.2. Refined Model and Hypothesis

Given the explorative results driven by in-depth interviews and EFA, we refined our conceptual process model and developed a hypothesis (see Figure 2).

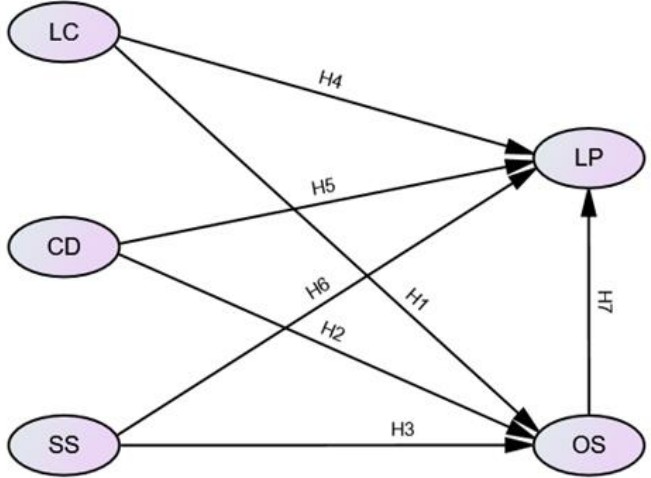

**Figure 2.** Refined model of the online-learning stress mechanism.

Firstly, we proposed the following hypotheses concerning the influential factors of online-learning stress:

**Hypothesis 1 (H1).** *Learner competence has a negative impact on online-learning stress.*

**Hypothesis 2 (H2).** *Course design reasonability has a negative impact on online-learning stress.*

**Hypothesis 3 (H3).** *Social support has a negative impact on online-learning stress.*

Secondly, we noticed in the interviews that even those influential factors could directly impact the learning performance of college students during the period of online learning. Therefore, the study developed the following hypotheses:

**Hypothesis 4 (H4).** *Learner competence has a positive impact on the learning performance of college students.*

**Hypothesis 5 (H5).** *Course design reasonability has a positive impact on the learning performance of college students.*

**Hypothesis 6 (H6).** *Social support has a positive impact on the learning performance of college students.*

Finally, we continued to believe that perceived online-learning stress could also directly change the learning performance of college students:

**Hypothesis 7 (H7).** *Online-learning stress of college students has a negative impact on their learning performance.*

### 3.2.3. CFA Test and Measurement Model

To confirm the quality of the online-learning stress mechanism's refined model, we applied CFA to test the reliability, convergent validity, and discriminant validity of the measurement model. To reach the model fit, we adjusted the measurement model by excluding two items from OS (OS4 and OS5) and connected the relationships among the influential factors (LC, CD, and SS).

Specifically, loadings and SMC should exceed 0.50 and 0.25, Cronbach's alpha coefficient and composite reliability (CR) should exceed 0.70, and the average variance extracted (AVE) of all factors should be above the value of 0.50. Apparently, the adjusted, refined model met all criteria of reliability and convergence validity in terms of the measurement model [30,31] (see Table 5).

**Table 5.** Reliability and convergence validity indices of the adjusted, refined model.

| Factor | Concept | Loadings | std. | Unstd. | S.E. | t-Value | *p* | SMC | CR | AVE |
|--------|---------|----------|------|--------|------|---------|-----|-----|-----|-----|
| LC | LTC | 0.64 | 0.802 | 1 | | | | 0.410 | | |
| | MC | 0.67 | 0.819 | 0.599 | 0.058 | 10.39 | *** | 0.449 | 0.830 | 0.620 |
| | SE | 0.54 | 0.738 | 0.483 | 0.054 | 8.902 | *** | 0.292 | | |
| CD | DC | 0.48 | 0.678 | 1 | | | | 0.230 | | |
| | CDR | 0.58 | 0.76 | 0.941 | 0.113 | 8.355 | *** | 0.336 | | |
| | WLR | 0.67 | 0.816 | 1.241 | 0.137 | 9.051 | *** | 0.449 | 0.870 | 0.574 |
| | TF | 0.65 | 0.808 | 1.205 | 0.14 | 8.601 | *** | 0.423 | | |
| | PPK | 0.51 | 0.717 | 1 | 0.128 | 7.836 | *** | 0.260 | | |
| SS | ES | 0.49 | 0.698 | 1 | | | | 0.240 | | |
| | TI | 0.82 | 0.906 | 1.39 | 0.143 | 9.735 | *** | 0.672 | 0.857 | 0.668 |
| | CI | 0.7 | 0.835 | 1.31 | 0.138 | 9.476 | *** | 0.490 | | |
| LP | AC | 0.64 | 0.8 | 1 | | | | 0.410 | | |
| | CC | 0.89 | 0.89 | 1.153 | 0.094 | 12.273 | *** | 0.792 | 0.876 | 0.703 |
| | AP | 0.82 | 0.822 | 1.148 | 0.104 | 11.065 | *** | 0.672 | | |
| OS | OS2 | 0.6 | 0.776 | 1.191 | 0.125 | 9.499 | *** | 0.360 | | |
| | OS3 | 0.46 | 0.681 | 1.047 | 0.126 | 8.318 | *** | 0.212 | | |
| | OS1 | 0.5 | 0.707 | 1 | | | | 0.250 | | |
| | OS6 | 0.63 | 0.794 | 1.274 | 0.133 | 9.599 | *** | 0.397 | 0.933 | 0.638 |
| | OS7 | 0.72 | 0.851 | 1.36 | 0.133 | 10.245 | *** | 0.518 | | |
| | OS8 | 0.73 | 0.856 | 1.376 | 0.133 | 10.356 | *** | 0.533 | | |
| | OS9 | 0.71 | 0.846 | 1.363 | 0.134 | 10.184 | *** | 0.504 | | |
| | OS10 | 0.73 | 0.856 | 1.345 | 0.13 | 10.307 | *** | 0.533 | | |

Note: *** *p* < 0.001.

We also confirmed the discriminant validity of each factor in the measurement model using Fornell–Larcker criterion [32]. As presented in Table 6, the square root of the AVE of each factor was greater than the absolute value of the Pearson correlation coefficient between the pairs of factors.

**Table 6.** Fornell-Larcker criterion for discriminant validity results of the adjusted, refined model.

| Factor | SS | LC | CD | OS | LP |
|--------|------|------|------|------|------|
| SS | 0.817 | | | | |
| LC | 0.486 | 0.787 | | | |
| CD | 0.669 | 0.432 | 0.758 | | |
| OS | −0.383 | −0.471 | −0.428 | 0.799 | |
| LP | 0.574 | 0.677 | 0.498 | −0.502 | 0.838 |

Finally, we tested the model fit of the adjusted, refined model. According to scholars, the value of χ2/df (degree of freedom) should range from 1 to 3, the value of GFI (goodness

of fit index) should be above 0.80, and the value of RMSEA (root-mean-square error of approximation) should be smaller than 0.08. Our model passed the examination of model fit with $\chi2$ = 382.264, df = 199, $\chi2$/df = 1.921, GFI = 0.827, RMSEA = 0.076 [33].

### 3.2.4. Path Analysis

After confirming the quality of the measurement model, we performed the path analysis to test our hypothesis. The confirmation of the path effects was divided into two steps. Firstly, we referred to the unstandardized regression weights. If the unstandardized factor loadings and path coefficients were significant ($p < 0.05$), the hypothesis could be supported. When a hypothesis was supported, we then implemented the second step: the evaluation of path effect strength based on standard regression weights. Scholars suggest using Cohen's effect size to test the impact of each path, with values between 0.02 and 0.15, between 0.15 and 0.35, and greater than 0.35, indicating low, moderate, and high effects, respectively [34]. The results of the path analysis are presented in Table 7 and Figure 3.

**Table 7.** Hypothesis testing results.

| Hypothesis | Relationship | Standardized Path Coefficient | *p* Value | Result |
|---|---|---|---|---|
| H1 | LC→OS | −0.339 | 0.001 | Supported |
| H2 | CD→OS | −0.245 | 0.039 | Supported |
| H3 | SS→OS | −0.055 | 0.645 | Not supported |
| H4 | LC→LP | 0.45 | 0.001 | Supported |
| H5 | CD→LP | 0.068 | 0.508 | Not supported |
| H6 | SS→LP | 0.246 | 0.024 | Supported |
| H7 | OS→LP | −0.167 | 0.042 | Supported |

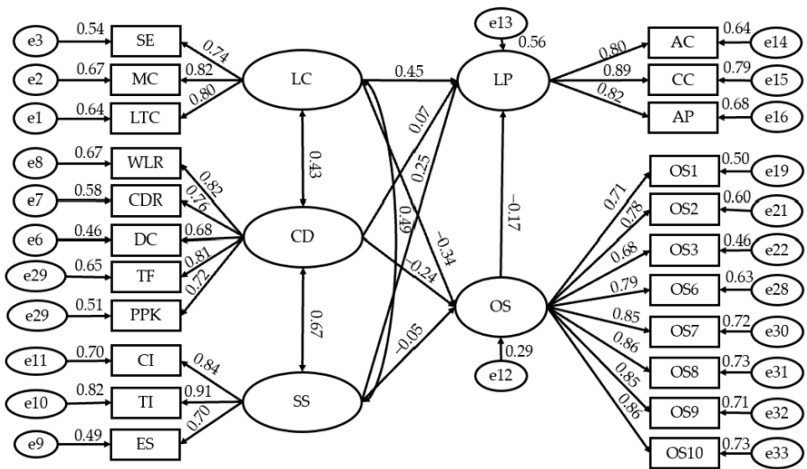

**Figure 3.** The adjusted, refined model of the online-learning stress mechanism.

The results from path analysis indicated that both LC and CD had a moderate negative impact on OS. LC had a large impact on LP, while SS only had a moderate effect on LP. The negative effect of OS on LP was also proved to be moderate.

Regarding the mediating effect, we could conclude that OS partially mediated the relationship of LC→OS→LP while fully mediating the path of LC→CD→LP in a negative way. OS did not work as a mediator in the path of SS→LP.

### 3.2.5. Final Confirmed Version of Model

Finally, we excluded the unsupported paths and retested the model. All the tests of validity, reliability, and model fit in terms of measurement model were passed. The results

regarding the path analysis also proved that the final confirmed version of the model could best describe the relationship between factors (see Figure 4).

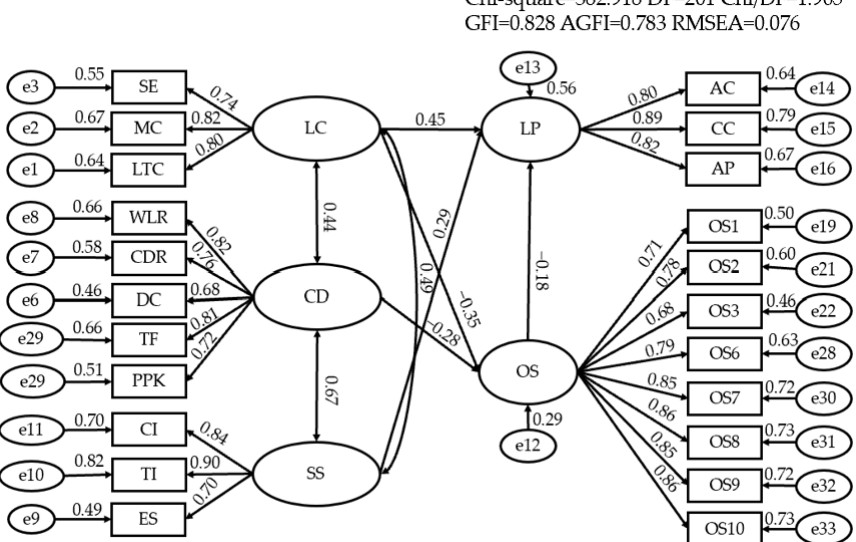

**Figure 4.** Final confirmed model of the online-learning stress mechanism.

## 4. Discussion

The study uses a combination of exploratory and confirmatory research methods to uncover the mechanism of college students' online-learning stress. The results from an exploratory qualitative study following the QUAGOL present altogether 24 key concepts in terms of online-learning stress, among which 11 are manifestations of online-learning stress, 10 are diverse antecedents of online-learning stress, and three are the study-relevant outcomes of online-learning stress. College students who participated in the interviews also expressed their understanding of the relationship between these three dimensions: first of all, the antecedents of online-learning stress can directly impact the level of online-learning stress perception. Then, the level of online-learning stress can also directly impact their learning performances. In addition, it is also possible that those key antecedents of online-learning stress can directly lead to changes in learning performances during the period of online learning.

Next, we explored and refined the mechanism of online-learning stress of college students by using the quantitative research method of SEM. We managed to subdivide 10 antecedents of online-learning stress into three main factors (learner competence and commitment, course design reasonability, and social support). Together with online-learning stress and learning performance, we finally formed a five-factor model that could best describe the mechanism of online-learning stress with the interaction among five factors according to our data collected from the online survey.

Based on the two-step methods (qualitative and quantitative) applied in our research, results are discussed in the following two parts.

### 4.1. Discussion on the Qualitative Study Results

We surprisingly noticed that the manifestations of online-learning stress are of great diversity. In general, online-learning stress can be defined as feelings of inability, unwillingness, irritation, and a lack of confidence toward online learning. Forms of online-learning stress vary based on emotional intensity, which refers to variations in the magnitude of emotional responses [35]. Specifically, manifestations of online-learning stress range from light discomfort of attention distraction to horrible feelings of total breakdown and inability to do anything in life. The emotion of online-learning stress can be either covert (hidden and concealed) or overt (blatant and obvious) [36,37]. In the interview,

some participants expressed their covert emotion of stress as tense and nervous while taking online courses without being noticed by others. Others, on the contrary, possessed overt emotions of online-learning stress: they become irritated and angry while working on online assignments to the extent that even their classmates and family members were easily aware of their stress. The diverse forms of online-learning stress added to the definition of online-learning stress proposed by existing literature [16,17].

The influential factors of online-learning stress are also complex. We find that the influencing factors vary from person to person, especially depending on their personal type of locus of control. Locus of control reveals how people explain their personal experience, or in other words, their attribution preferences [38]. Rotter divides the concept of locus of control into two dimensions: internal and external. An internal locus of control attributes the outcome normally to him/herself, whereas an external locus of control prefers to attribute the outcome to external factors such as chance, other people, or luck [39]. In our study, participants can also be divided into internals and externals in terms of the locus of control: some participants tend to attribute the online-learning stress to their weak personalities and blame themselves at the very beginning of the interview, while others, on the contrary, look for external or environmental factors first. They attribute their feeling of stress to problems, such as the irrationality of course design, poor competence, the attitude of teachers, technical problems, etc. Our findings that different types of personality lead to different influential factors of online-learning stress are also supported by other scholars [40,41]. Another interesting finding is that the effects of those influential factors on online-learning stress are not consistent. In the study, different participants sometimes have totally opposite views on the impact of a specific influential factor on online-learning stress. The most controversial influential factor is the interaction with teachers. It can contribute to stress for some participants but relieve others. To the best of our knowledge, there is no discussion on the controversy of influential factors in the existing literature so far.

The outcomes presented in the qualitative study results are generally in line with most of the arguments in previous studies [23]. However, we identify more outcomes in the study. Participants list several outcomes during the stressful online-learning period, ranging from emotional and attitudinal changes across cognitive and comprehensive changes to behavioral changes. Controversy changes in terms of attitude, cognition, and behavior during the online learning period are also argued by different participants. This finding echoes the arguments of scholars who think positively of stress [17,24].

We also find that though many participants mention a change in their learning performance after feeling stressed with online learning, many of them unconsciously link antecedents directly to outcomes. For example, some admit that the decrease in their commitment to online learning is the main reason for their poor learning performance. The direct path of influential factors (self-efficacy, time commitment, teacher-student interaction, etc.) to online-learning performance also holds in recent studies [42–44]. In this way, the mechanism of online-learning stress is further expanded by both the direct and indirect role of influential factors.

### 4.2. Discussion on the Quantitative Study Results

The first result of the quantitative study is that 10 antecedents of online-learning stress can be further categorized into three main factors. We reached the results of three main factors from a purely inductive and explorative research progress with the help of EFA. According to the rotated factor matrix, we generalize DC, WLR, CDR, TF, and PPK into the factor of CD. In addition, CI, TI, and ES are classified into the factor of SS, and SE, LTC, and MC are grouped into the factor of LC. Thus, the model of the online-learning stress mechanism could be refined into a five-factor model. These influential factors (key concepts) are especially in line with some previous findings [20,21]. However, our findings outperform the previous findings by using both qualitative and quantitative research methods to explore and confirm the classification of influential factors.

The second result of the quantitative study concerns the test of the hypotheses among factors with the tool of CFA and SEM-path analysis. We found that LC (SE, LTC, and MC) and CD (DC, WLR, CDR, TF, and PPK) have a moderate negative impact on OS, supporting H1 and H2. This means that the lower the level of course design reasonability, competence, and commitment of the learner, the more stress they perceive. When we compare the effect size of course design reasonability and learner competence and commitment, we find that the latter is even more important than the former for perceived stress. SS (CI, TI, and ES) fails to have a significant effect on OS, rejecting H3. As we discussed in the first section, social support is listed as an important antecedent for decreasing the level of online-learning stress in some articles [19,22]. However, according to our study, this effect is in doubt. When we recall the interviews in the qualitative study, this result would be more understandable since some regard this kind of social support online as formalistic and meaningless and may lead to deeper feelings of stress. This result is likely to be closely related to the cultural context of the study, i.e., the unique teacher–student relationship in China. Sometimes, the support from Chinese teachers is recognized as "autonomy support", which refers to offering students choices and encouraging students to develop autonomously. However, many Chinese students do not appreciate this form of support [45]. The results of several studies in the Chinese context are also in line with our study [46]. In any event, OS is confirmed to have a moderate negative effect on LP, supporting the H7. This result implies that the more pressure students feel, the worse their learning performance will be. Therefore, the logical path of influential factors across online stress to learning performance is proven to be valid. Meanwhile, LC and SS can also improve LP directly, supporting H4 and H6. Therefore, learner competence and social support while online learning is confirmed to be vital for college students' learning success. The former is more important than the latter concerning its effect on learning performance based on their effect sizes.

The third result of the quantitative study is the negative mediation effect, or the so-called suppression effect in our case of OS, in the whole mechanism. Findings support that OS works as a negative partial mediator for LC, i.e., the positive effect of LC on LP can be suppressed by the online-learning stress perceived by college students. We also unexpectedly identified the full mediation effect of OS on the path of CD to LP. In other words, the course design reasonability cannot directly improve the learning performance of college students in the period of online learning without decreasing online-learning stress. To the best of our knowledge, a similar research theme has not been discussed in previous studies.

## 5. Conclusions

The findings from both qualitative and quantitative studies provide a vivid picture of college students' stress-related experiences during the online-learning period: since the outbreak of COVID-19, students have constantly been dealing with all kinds of potentially influential factors that could impact their level of online-learning stress. Some factors are more influential to stress, while others may have less impact. As an outcome of stress, their learning performance may be worse than before.

Our study provides both theoretical and practical implications. As for the theoretical implications, we proposed a rigorous theoretical research framework, namely combining both qualitative (in-depth interviews and QUAGOL analysis) and quantitative research methods (online survey and SEM analysis) to build up a solid conceptual process model. In addition, the study enriches the forms of online-learning stress of college students. This also contributes to the definition of online-learning stress. Moreover, we connect the antecedents and outcomes of the online-learning stress and thus uncover the general mechanism of online-learning stress which has rarely been studied by former scholars.

As for the practical implications, our findings could help both sides of online learning and teaching. According to our results, students' competence and commitment are important to the level of stress. For example, students are recommended to commit even more time to study than offline learning. Students are recommended to apply specific time

management apps to improve concentration and increase the length of their studies. On the side of teaching, our findings suggest that teachers be more careful with the difficulties and workload of the course. If possible, teachers should choose the online-teaching platforms that enable students to make good use of multiple devices. Teachers could moderately increase the proportion of teaching professional knowledge compared to interaction sessions with students since online courses are more demanding on their professionalism and expertise rather than their social support and interaction with students. Furthermore, our findings could also benefit higher education institutions in a broader way. Instead of building up the digital platform of social support, higher education institutions should pay more attention to the capacity building of students and the systematic optimization of online learning and teaching platforms. More specifically, courses contributing to learning competence enhancement could be covered in the regular curriculum. In addition, the system of online learning and teaching within higher education should be better equipped with functions to control the quality of online courses at all times.

Our study also has its limitations and requires further research in the future. Firstly, our sample is still limited to Chinese college students to exclude cultural impacts. However, further studies could also test our proposed model by expanding the context of cross-cultural scenarios. For example, future studies could explore if the effect of the model in countries with lower uncertainty avoidance is stronger compared to that in China. In addition, our sampling process is not able to take the diverse majors of the participants, locations, and number of the participants' colleges into account, which may also affect the results concerning online-learning stress. Thus, future scholars could continuously work on the inter-group diversity among college students in terms of the mechanism of online-learning stress. All of our data are subjective self-reports from participants who recall and express their feelings during the period of online learning. Participants may forget, modify, or even lie during the self-reports, resulting in data that does not fully reflect reality. Therefore, in future studies, scholars could improve the objectivity of data through the collection of second-hand social media data or observation. Lastly, we noticed from the in-depth interviews that some college students also report a certain level of stress after switching back to offline learning. Together with offline-learning stress and online-learning stress, this "reverse offline-learning stress" enriches the types of learning stress and completes the "learning stress cycle" of college students. How students constantly deal with specific forms of stress at different stages is a critical issue for educational sustainability that deserves future research.

**Author Contributions:** Conceptualization E.W.; methodology, X.Z.; validation, E.W.; formal analysis, E.W.; investigation, X.Z.; writing—original draft preparation, E.W.; writing—review and editing, E.W. and X.Z.; visualization, X.Z.; supervision, E.W. project administration, funding acquisition, All authors have read and agreed to the published version of the manuscript.

**Funding:** This research was supported by the 2021 Shanghai Education Science Research Project "Exploring the Cultivation Model of Foreign Language Talents under the Perspective of "New Liberal Arts"-Research on the Development of Cross-cultural Competence" Project No. C2021255.

**Institutional Review Board Statement:** Not applicable.

**Informed Consent Statement:** Not applicable.

**Data Availability Statement:** Not applicable.

**Conflicts of Interest:** The authors declare no conflict of interest.

## Abbreviation

| Abbreviation | Meaning |
|---|---|
| OS | online-learning stress |
| LP | learning performance outcomes |
| LC | learner competence and commitment |
| CD | course design reasonability |
| SS | social support |
| OS1 | Feeling of attention distraction |
| OS2 | Feeling of an inability to control important things in learning |
| OS3 | Feeling of mental tension for no reason |
| OS4 | Feeling of losing confidence in dealing with personal issues |
| OS5 | Feeling of unwilling direction in which things are going |
| OS6 | Feeling of an inability to follow the daily routine |
| OS7 | Feeling of an inability to tackle the irritating things in studies |
| OS8 | Feeling of an inability to control anything in life |
| OS9 | Feeling of irritation and anger towards studying |
| OS10 | Feeling of an inability to complete multiple tasks |
| CI | Interaction with classmates |
| TI | Interaction with teachers |
| ES | Emotional support during course |
| SE | Self-efficacy |
| LTC | Learning time commitment |
| MC | Accessibility to combined multiple learning methods |
| DC | Accessibility to combined online-learning devices |
| WLR | Reasonability of workload |
| CDR | Reasonability of course difficulty |
| TF | Timely feedback by the teacher of the course |
| PPK | Teacher's proficiency in professional knowledge of the course |
| AC | Attitudes changes |
| CC | Cognitive changes |
| AP | Academic performance changes |

## Appendix A

| Question | Dimension |
|---|---|
| Do you have any experience in taking online learning? When did you start taking online learning? | Background information |
| Do you think the technology required to take online courses is difficult? Which learning platform is the most popular one? | Background information |
| What benefits do you get from online learning compared to offline courses? In what ways? | Background information |
| Does online learning cause you any feelings of stress? If so, how much? In what ways? | Online-learning stress manifestations |
| What are the sources for your feeling of stress, in your opinion? | Influential factors of online-learning stress |
| Compared to offline learning, do you think you spend more or less time on learning? | Influential factors of online-learning stress |
| Did you have communication with your classmates and teacher during the online classes? | Influential factors of online-learning stress |
| How did the experience of online learning affect you? Did the experience of online-learning stress have both positive and negative impacts on your learning performance? | Impact of online-learning stress |
| How would you expect the online-learning in the future? | Future expectation |

### Appendix B

| No. | Question | Dimension |
|---|---|---|
| 1 | Gender | |
| 2 | Grade | Demographic background |
| 3 | Major | |
| 4 | In general, I am confident in my ability to learn, and I usually have full control over my learning behavior. | |
| 5 | I devote sufficient study time to online courses. | |
| 6 | When I study online, I combine offline learning methods before, during, and after class. | |
| 7 | I have a variety of learning devices to support me when I take online classes. | |
| 8 | I think the workload in online courses is reasonable in most cases. | Influential factors of online-learning stress |
| 9 | I think the difficulty of the online courses is moderate in most cases. | |
| 10 | My teacher is able to provide timely feedback on the questions I ask. | |
| 11 | My teacher has a satisfactory level of teaching expertise. | |
| 12 | I think that the online courses allow for adequate interaction with classmates. | |
| 13 | I think that online classes allow for adequate interaction with the teacher. | |
| 14 | I can often feel the emotional support when I take online courses. | |
| 15 | How many times have you been distracted by unexpected things while studying online? | |
| 16 | How many times have you felt out of control of important things in your life while studying online? | |
| 17 | How often do you feel tense and stressed when studying online? | |
| 18 | How often do you feel not confident in your ability to handle personal problems when studying online? | |
| 19 | When studying online, how often do you feel that things are not going as you want them to? | |
| 20 | When studying online, how often do you find that you cannot cope with all the things you have to do? | Online-learning stress manifestations |
| 21 | How many times have you been able to control the irritating aspects of your learning while studying online? | |
| 22 | How many times have you felt like you had everything under control when studying online? | |
| 23 | How often do you get angry when you study online because something happens that you cannot control? | |
| 24 | How many times have you felt like difficulties were piling up and you couldn't overcome them while studying online? | |
| 25 | My attitude and evaluation of the course changed for the better after taking the online courses. | |
| 26 | My understanding and mastery of the course content strengthened after taking the online courses. | Impact of online-learning stress |
| 27 | After online learning, I achieved satisfactory examination results. | |

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
