# Peer review of "Uncovering the Mechanism of Online-Learning Stress of College Students"

_sustainability, doi:10.3390/su15129541_

Round 1

Reviewer 1 Report

The manuscript reports the factors of online-learning stress, the manifestations of online-learning stress, and the impacts of online-learning stress on learning performance. Through qualitative and quantitative research with college students, the study found that the factors of online-learning stress could be broken down into three sub-dimensions: learner competence and commitment (LC), course design reasonability (CD), and social support (SS). The process model tested and refined through qualitative and quantitative analyses can benefit educators preparing online-learning courses in higher education, as well as furnishing adult education through virtual training, technical assistance, and coaching.

The manuscript is well-structured and well-written. There are a few suggestions that could help make the manuscript even stronger, especially in the methods section.

Methods

Sampling of Online Survey (page 4, line 131-136): The author(s) generally describes participants as college students; however, to provide a more in-depth description of the sample, the number of colleges/universities involved, their geographical location (e.g., rural and urban), and majors of the participants (e.g., Arts, Statistics, and so on) should be included. Additionally, the major and grade of the participants should be noted in the survey to further enable the audience of the article to have a better understanding of the sample. The number and location of college(s) will also be applied to the in-depth interview participants.

The authors did not describe the sampling procedure. Therefore, it is recommended that the author(s) should add a separate section, titled either 'Sampling' or 'Participants', to describe the recruitment and demographic information of the participants for both qualitative and quantitative studies.

Qualitative Data Analysis (page 4, line 140-144): It would be beneficial to have a more thorough understanding of the qualitative data analysis process. Specifically, it is important to know who conducted the analysis: was it multiple trained researchers or a single analyst? Furthermore, as QUAOL has 10 consecutive stages of the coding process, it would be helpful to know the extent to which these stages were fully followed (or just first stage of them were followed) during the analysis.

Discussion & Conclusion

It is intriguing to observe that the interaction with teachers can add to the stress of some participants, and there appears to be no effect of Social Support (SS) in reducing the stresses associated with online-learning. It could be connected to a distinct cultural context that is unique to China; as the authors have highlighted, being limited to a single cultural context is an issue. It would be interesting to see if there are any findings in other literature regarding the unique teacher-student relationship in China that could be applied to this situation, though any over-interpretation should be avoided.

The implications of these findings can be further explored to provide more practical suggestions and viable strategies for higher education administration. Rather than focusing on individual students or teachers, attention should be given to building systems within higher-education institutions that are better equipped to facilitate online learning. It would be highly beneficial if higher-education institutes can place greater emphasis on capacity-building with the findings in the manuscript.

Author Response

Dear reviewer,

Thank you for your professional review on our article! As you are concerned, there are several problems that need to be addressed. According to your nice suggestions, we have made extensive corrections to our previous draft, the detailed corrections are listed below:

  • Sampling of Online Survey

As you kindly suggested, we completed the demographic information of all the online survey participants, including their gender, major, grade, and online-learning experience (page 5, Figure 1).

Besides, as for the sampling procedure, we finally decided to introduce our sampling methods separately in section 2.1. In-depth interview (page 3, line 127-130) and section 2.2. Online survey (page 4, line 161-164)

  • Qualitative Data Analysis

We added more details to present the process of qualitative data analysis by confirming that we went through all the 10 stages (page 5, line 182) and providing information about three analysts who coded and confirmed the coding process (page 5, line 185-188). 

  • Discussion on no effect of SS on OS

We really appreciated your idea of linking the no effect of SS to the specific cultural context of Chinese higher education! Thus, we went through the relevant literature and further discussed the result in the Chinese context (Page 17, line 594-599).

  • Practical implication for higher education institutions

We agree that higher education institutions should play a role in reducing the online-learning stress of college students thanks to your kind reminder. Thus, we added more sentences to imply possible measures taken by the higher education institutions from a broader and more systematic view (page 18, line 645-652).

We tried our best to improve the manuscript and made some changes marked in the revised paper. We appreciate your warm work earnestly, and hope the correction will meet with approval.

Once again, thank you very much for your comments and suggestions.

Best regards,

Enuo Wang

Xueyao Zhang

Reviewer 2 Report

You have written a quality paper but you need to address some minor errors to improve the standard of the paper. I explain my concerns in more detail below. I ask that the authors specifically address each of my comments in their responses.

1.     Which sample method was used to select college students for the interview? Also, did you take the ethics committee the approval of regarding the interview questions? It is very important for the scientific papers.

2.     How did you select (sample method) the participants for the Perceived Stress Scale? Also, demographic information about the participants should be given.

3.     If dimensions are also given in APPENDIX A, the scale is better understood by the readers

4.     If dimensions are also given in APPENDIX B, the scale is better understood by the readers.

The quality of the English Language of the paper is satisfactory.

Author Response

Dear reviewer,

Thank you for your professional review of our article! As you are concerned, there are still some problems that we need to solve and improve. According to your suggestions, we have made extensive corrections to our previous draft, the detailed corrections are listed below:

  • Sampling of Online Survey

As you kindly suggested, we introduced our sample method in section 2.1. In-depth interview (page 3, line127-130). Also, we added a sentence introducing the sample method of online survey in section 2.2. Online survey (page 4, line 161-164).

  • Ethical issues of interview and survey

We did have the approval in terms of our interview and survey questions from the ethic committee of our home university. Following your suggestion, we added more details about the ethical issues in both sections of data collection (page 4, line 141-147 & page 5, line 169-173).

  • Dimensions within Appendix

Both Appendix A and Appendix B were now added with dimensions for better understanding (page 20 & page 21).

We tried our best to improve the manuscript and all the changes were marked in the revised paper. We appreciate your kind revision, and hope our correction will meet with approval.

Once again, thank you very much for your comments and suggestions.

Best regards,

Enuo Wang

Xueyao Zhang

Reviewer 3 Report

The paper titled "Uncovering the mechanism of online-learning stress of college students" addresses an important issue in higher education regarding the sustainability of online learning. While the study's objective and methodology are commendable, there are several areas where the paper could benefit from improvement.

Firstly, the paper's overall length is excessive and lacks proper organization and structure. The information presented in the abstract and introduction should be concise and focused, providing a clear overview of the study's purpose, methodology, and key findings. Streamlining the content will enhance readability and allow readers to grasp the main points more effectively.

Furthermore, the discussion section requires reorganization to better convey the ideas and support them with recent works from high-level literature. By citing relevant studies and integrating their findings into the discussion, the authors can establish a stronger theoretical framework and demonstrate the paper's contribution to the field. This will also provide readers with a broader context for understanding the implications of the study's results.

Additionally, it is important to highlight the need for more recent references in the paper. Including up-to-date literature from reputable sources will strengthen the study's validity and ensure that the findings are situated within the current scholarly discourse. This will also demonstrate the authors' engagement with the latest research developments in the field of online learning stress.

In conclusion, while the paper addresses an important topic and utilizes a mixed methods approach, it would greatly benefit from improvements in its organization and structure. Streamlining the content, reorganizing the discussion section, and incorporating recent literature will enhance the clarity, relevance, and impact of the study. These revisions will strengthen the theoretical underpinnings of the research and contribute to the advancement of knowledge in the field of online learning stress.

Finally, please see specific comments in the attached file.

Author Response

Dear reviewer,

Thank you for your professional review on our article! As you are concerned, there are several problems that need to be addressed. According to your inspiring suggestions, we have made extensive modifications to our previous manuscript. In this revised version, changes to our manuscript were all highlighted within the document by using red-colored text. Point-by-point responses to your suggestions are listed below:

  • Reorganization of the abstract and introduction

As you kindly suggested, we modified our abstract (page 1, line 14-18), and revised extensively with the paragraphs within introduction. During the revision, we deleted redundant contents and rewrote the paragraphs according to your comments in the PDF version (see all the red parts on page 1-3, line 23-111).

  • Ethical issues of interview and survey

We did have the approval in terms of our interview and survey questions from the ethic committee of our home university. Following your suggestion, we added more details about the ethical issues in both sections of data collection (page 4, line 141-147 & page 5, line 169-173).

  • A brief introductory sentence that outlines the purpose of presenting the qualitative study results

We added the sentence as your suggestion to introduce the purpose of presenting qualitative study results (page 5, line 197-199).

  • A thorough discussion of the results based on high-level and recent literature

Thanks to your comments, we noticed that our former paper failed to conduct a solid and persuasive discussion. We rewrote most of the paragraphs following your professional comments. We clarified the forms of online-learning stress based on literature of emotional intensity and demonstrated our findings with the examples in our interview (page 15, line 502-512). As for the influential factors of online-learning stress, we referred to high-level literature of the theory locus of control, and compared our findings with the findings derived by other scholars (page 15, line 518-532). We also tried our best to discuss both the direct and the indirect role of the influential factors in-depth by relating our findings to the extant findings (page 16, line 555-560).

  • More concrete practical implications

We added up more specific and practical implications for both students (page 17, line 634-635) and teachers (page 17-18, line 642-652)

  • More concrete limitations and future research

We edited extensively with the section of limitation and future expectation according to your suggestions. We expanded all the statements of limitations and future expectations by giving examples or detailed clarifications (page 18, line 655-670).

  • More recent references in the paper

We additionally cited 19 more articles in our revised manuscript, among which 14 articles were released in recent three years (page 24-25).

We tried our best to improve the manuscript following all your comments. We appreciate your professional work earnestly, and hope the correction will meet with approval.

Once again, thank you very much for your comments and suggestions.

Best regards,

Enuo Wang

Xueyao Zhang

Reviewer 4 Report

Congratulations, you have successfully submitted an article to this journal. Overall the contents of your article are very good, but there are some parts that we have written suggestions for revision. Please take a look at the PDF version of the article provided to us, we hope that your article can be received and published as soon as possible. We apologize for our lack of ability to provide input on your article.

Author Response

Dear reviewer,

We appreciate your professional review on our manuscript! As you suggested, there are still some problems that we need to deal with during the revision. According to your suggestions, we have made extensive corrections to our previous draft, the detailed corrections are listed below:

  • Formulation of the first research question

We all agreed with your comments that the formulation of the first research question is not persuasive and reasonable enough since it’s too theoretical in nature. Based on your comments, we realized that what we tried hard to explore in our research is not the definition of the online-learning stress, but the manifestations of the online-learning stress. Manifestation of the online-learning stress is also an important dimension within the online-learning mechanism, which is rarely discussed in previous literature. Thus, we would like to change our first research question from “what is online-learning stress” to “what are the manifestations of online-learning stress” (page 2, line 89). We hope this change could be acceptable to you.

  • Data Analysis

We added more details to present the process of data analysis by confirming the stages we went through (page 5, line 182) and providing information about three analysts who coded and confirmed the qualitative coding process (page 5, line185-188). We also added the name of the software we used to conduct quantitative data analysis (page5, line 191).

  • Appendix for abbreviations

We really appreciated your kind suggestion of adding an appendix page that explains what each indicator stands for in the study. Following your guidance, we added Appendix C to the manuscript including all the concepts that appeared in our article (page 23).

We tried our best to improve the manuscript and all the changes were marked in red in the revised paper. We appreciate your kind revision, and hope our correction will meet with approval.

Once again, thank you very much for your comments and suggestions.

Best regards,

Enuo Wang

Xueyao Zhang

Round 2

Reviewer 1 Report

The manuscript has been improved by incorporating the recommendations. I am grateful for the demographic information from the online survey and the sampling method information. Nevertheless, there are a few more suggestions that can further refine the manuscript before its publication.

1. More description concerning sampling was suggested, but I still don't feel like it gives enough detail.

I suggest the author(s) include the following additional information in the 'Sampling' (or 'Participants') section of the manuscript, in addition to all other information regarding participants.

For the In-depth Inteview:

If a purposive sampling method is employed for conducting in-depth interviews, the author(s) can provide information about the following questions.

  • How many colleges or universities are the 15 participants from?

  • What are geographical location of those colleges or universities?

For the Online Survey:

  • What is the process of the ‘Snowball Sampling’?

    • How many participants were recruited to serve as the initial seeds?

      • How many colleges or universities were involved at the initial recruitment?

      • What are geographical location of those colleges or universities at the initial recruitment?

    • How the further recruited was conducted from there?

2. I recommend the authors provide a table for participant demographic information instead of pie charts. It is difficult to differentiate the proportions of the participants' majors based on the color in the pie chart.

3. It is suggested to include a mention of potential limitations caused by the sampling. For example, the major of the participants, the locations (e.g., urban/rural, wealthy/poor), and the number of colleges included were not taken into account, which may affect perceptions and stress levels related to online learning.

Author Response

Dear reviewer,

Thank you again for your review of our article! As you are concerned, there are still some problems that need to be addressed. According to your professional suggestions, we have made extensive corrections to our manuscript. Please kindly check the detailed corrections listed below:

  • Sample or participant section

As you suggested, we added a section named “Participants” introducing all the necessary information about our participants and the process of how they were selected (page 3-5, line 123-144). We separately clarified the sample and sampling process of in-depth interview and online survey.

As for the in-depth interview, we further presented information about colleges and their locations of our participants (page 3, line 124-134). We really regret that we were unable to show the exact name of the college and city of our participants any further, as some students were concerned that they would be negatively influenced by making comments about their experience of online-learning in their home college. We beg for your understanding and hope that the current level of disclosure is acceptable.

Following your inspiring comments, we also provided more information about the participants of our online survey and their selection process (page 4, line 135-144). More specifically, we introduced the exact name of the social media platform where we recruited 43 initial participants and explained why we recruited on this social media platform as well. Besides, we also informed our potential readers about the locations and number of colleges of our participants in the online survey.

Moreover, your suggestion on the presentation form of demographic information was much appreciated. Thus, we replaced pie charts with a table to present demographic information of online survey participants (page 4, Table 2).

  • Limitation of sampling

Your suggestion on the limitations of the sampling process was enlightening. So, in the latest version of manuscript, we took this limitation into consideration and also expected future discussions on demographic diversity and its impacts on the mechanism of college students’ online-learning stress (page 19, line 690-694).

We tried our best to improve the manuscript. We appreciate your warm work earnestly, and hope the correction will meet with approval.

Once again, thank you very much for your comments and suggestions.

Best regards,

Enuo Wang

Xueyao Zhang

Reviewer 3 Report

Dear authors,

I would like to express my gratitude for considering and incorporating my revisions in the manuscript. The paper has undergone significant improvements, and I am pleased with the progress that has been made.

The objective of this study is to address the challenge of online-learning stress in higher education and propose a conceptual process model that elucidates the mechanism of online-learning stress among college students. The study utilizes a mixed methods approach, combining qualitative and quantitative methods to provide a comprehensive understanding of online-learning stress.

The qualitative study involved in-depth interviews with 15 college students, resulting in the identification of 11 influential factors of online-learning stress, as well as ten manifestations of online-learning stress (OS) and three learning performance outcomes of OS (LP). These findings contribute to the development of a robust conceptual framework.

Additionally, the quantitative study involved 159 online surveys, which allowed for the further categorization of influential factors into three sub-dimensions: learner competence and commitment (LC), course design reasonability (CD), and social support (SS). The results of a structural equation model (SEM) support the negative impact of LC and CD on OS, as well as the negative impact of OS on LP. However, the hypothesized negative effect of SS on OS did not receive empirical support.

The study not only contributes to the development of online-learning stress theory but also has practical implications for online learning and teaching in higher education. By gaining insights into the influential factors and manifestations of online-learning stress, educators and institutions can develop strategies to mitigate and manage online-learning stress, thereby enhancing the overall learning experience and performance of college students.

Once again, I commend the authors for their efforts in improving the manuscript. The revisions have enhanced the clarity and quality of the paper, making it a valuable contribution to the field. Therefore, I wholeheartedly endorse the publication of this study.

Sincerely,

Author Response

Dear reviewer,

Thank you again for your comment on our article!

We are grateful for your approval of the revised version. It is because of your detailed comments and suggestions in each section that we have been able to discuss online learning stress of college students in greater depth.

Once again, thank you very much for your comments, and I wish you the best of luck with your research!

Best regards,

Enuo Wang

Xueyao Zhang